# School Feeding to Improve Cognitive Performance in Disadvantaged Children: A 3-Arm Parallel Controlled Trial in Northwest Pakistan

**DOI:** 10.3390/nu15071768

**Published:** 2023-04-04

**Authors:** Nicola M. Lowe, Pamela Qualter, Jonathan K. Sinclair, Swarnim Gupta, Mukhtiar Zaman

**Affiliations:** 1Centre for Global Development, University of Central Lancashire, Preston PR1 2HE, UK; 2Institute of Education, School of Environment, Education and Development, University of Manchester, Manchester M13 9PL, UK; 3Rehman Medical Institute, Peshawar 25000, Pakistan

**Keywords:** cognitive performance, school feeding programme, Pakistan

## Abstract

Malnutrition is associated with reduced learning aptitude and growth during childhood. We examined the impact of providing two school lunch variants, a standard school meal (school feeding, *n* = 70), or the standard meal with additional micronutrients (school feeding + micronutrient powder (MNP), *n* = 70), in children attending two schools in northwest Pakistan. A third local government school, where no lunch was provided (no school feeding, *n* = 70), served as the control. The primary outcome, cognitive function, was assessed using the Raven’s Coloured Progressive Matrices (RCPM) test, alongside haemoglobin, at three-time points: T1 (baseline, before the initiation of the school lunch programme), T2 and T3 (5 and 12 months, respectively, after the introduction of the school lunch). Data were analysed using linear mixed-effects models to contrast between trial groups, the changes from T1 to T2 and T3. Adjusted for T1 and other co-variates, improvements in the RCPM scores were significantly greater in the school feeding group at T2 (b = 1.61, (95% CI = 0.71–2.52), t = 3.52, *p* = 0.001) and T3 (b = 1.28, (95% CI = 0.22–2.35), t = 2.38, *p* = 0.019) compared with no school feeding. In addition, at T2 (b = 1.63, (95% CI = −0.10–3.37), t = 1.86, *p* = 0.065), there were no significant differences between school feeding + MNP and no school feeding groups. However, improvements in the RCPM scores were significantly greater in the school feeding + MNP group at T3 (b = 2.35, (95% CI = 0.51–4.20), t = 2.53, *p* = 0.013) compared with no school feeding. The findings indicate an improvement in cognitive performance in children who received a school meal with and without MNP, over a 12-month period. Currently there is no operational school feeding programme at the national or provincial level in Pakistan. Our findings, therefore, highlight the need for school feeding programmes to improve learning opportunities for children from underprivileged communities.

## 1. Introduction

Addressing malnutrition continues to be an undisputed challenge, especially in the low- and middle-income countries (LMICs). Almost two hundred and fifty million children in LMICs are at risk of not attaining their full developmental potential because of malnutrition and poverty alone [1]. Poor development can result in reduced learning aptitude and academic achievements, poor health and productivity, and is associated with lower adult earning abilities [2,3,4]. These consequences not only affect the individual, but also the society by potentially endorsing and entrenching disparity and contributing to the intergenerational cycle of poverty.

In addition to inadequacy of macronutrients, micronutrient deficiencies, often referred to as ‘hidden hunger’, can have a negative impact on the growth and development of children. Micronutrients, including iron, zinc, iodine and copper, play essential roles in cognitive function, such as motor skills, memory and executive function [5]. Importantly, a meta-analysis has shown that micronutrient inventions can mediate consistent improvements in fluid intelligence among micronutrient-deficient children [5]. 

Pakistan, with a Global Hunger Index score of 26.1 (severity of hunger designated “serious”) and ranked 99/121 in the Global Hunger Index, is among the countries that are most affected by child malnutrition [6]. This index is calculated from national data on malnourishment, including child stunting (low height for age), child wasting (low weight for height) and child mortality, and provides a means of tracking whether a country is achieving hunger-related Sustainable Development Goals [7]. The most recent National Nutrition Survey (NNS) data show about 40.0% of children to be stunted and 15.0% wasted in the Khyber Pakhtunkhwa (KP) region (northwest) region of Pakistan [8]. The spread of vitamin and mineral deficiencies is common in the region, with 33.2% of children estimated to be deficient in iron, 18.6% in zinc and nearly half (46.7%) in vitamin A. The situation can often be worse in very marginalized communities. 

Nutrition interventions can support vulnerable children throughout their growth and development. With realization that malnutrition and infection remain the key constraints on development in children and mortality rates higher than previously understood, the value of investments beyond the first 1000 days of life are increasingly being recognized [9]. School feeding programmes (SFP) are increasingly being considered in LMICs to expand access to education and improve the nutrition and scholastic performance of children. However, the specific benefits of these interventions remain unclear [10,11]. Such programmes intend to reduce the short-term hunger that negatively impacts concentration span and learning capacity of school children, promoting gender equality and increasing enrolment, whilst also boosting community or family income [12,13]. 

Increasing micronutrient intakes can be achieved through the provision of supplements, or through fortifying food, either at the point of production (for example iodine added to salt) or through the addition of micronutrient powders to food before it is served [14]. Previous studies have shown this strategy to be successful in LMIC settings [15,16,17], however, there are significant cost implications which impact on the long-term sustainability of this approach. Providing additional meals that include animal source food (ASF) may be a more sustainable way to improve micronutrient intake [18,19].

The aim of this study was to assess the impact of providing a school meal compared with no school meal, on cognitive function in early-stage primary school children from a very low resource setting in northwest Pakistan. The potential additional benefit for cognitive function and iron status (haemoglobin levels) of fortifying the school lunch with micronutrients, in the form of a multiple-micronutrient powder (MNP), was also explored.

## 2. Materials and Methods

The study was conducted in a very low resource setting, where families live among the brick-kilns on the outskirts of Peshawar, Khyber Pakhtunkhwa Province (KPK). The study was undertaken in collaboration with a local non-government organization, the Abaseen Foundation Pakistan (AFPK), supported by a UK-based sister organization (AFUK). The community has a population of approximately 25,000, living in basic conditions, without household running water or sanitation. Meals are cooked outside or in a partially covered cooking area using solid fuel, and the houses are not connected to an electricity grid. Typically, children in this community consume a vegetable-based diet, comprised of bread (chapati, roti, paratha) and vegetable curry, served in the morning and evening. Most adult males and male children work on the brick kilns on a subsistence wage. According to the most recent census (2017) [20], the overall literacy rate in the KP province for people over 10 years of age is 69.2% for males and 38.7% for females, however, the literacy rate for both is likely to be lower in the brick-kiln communities, particularly for females. The AFPK operates two schools in the region, alongside government-provided primary schools. Typically, the school day starts at 8 am and ends at 2 pm. No meals were provided by any of the schools prior to the commencement of this study. In November 2015, AFPK introduced a school lunch programme at the two schools in this community (subsequently referred to as School A and school B). This provided an opportunity to investigate the impact of providing a regular meal for the children on their performance at school, in comparison to a local government school that served as a control (subsequently referred to as school C). 

### 2.1. Ethics and Trial Registration

Approval from the Khyber Medical University Committee and corresponding approval by the University of Central Lancashire (Unique Reference Number: STEMH 436) were obtained and the trial was prospectively registered (NCT05198024).

### 2.2. Sample Size

Power calculations were performed for the primary outcome variable, i.e., the between-groups magnitude of the change in Raven’s Coloured Progressive Matrices (RCPM). This showed that to provide α = 5% and β = 0.80, a total sample size of 210 was necessary to detect a change of 2.1 between the groups, with a projected standard deviation of 4.1, accounting for the loss to follow-up rate of 20%.

### 2.3. Recruitment

A letter was sent home with all the children in the preparatory class and class 1 in each of the three participating schools, to explain the purpose of the study, and informed consent was obtained from the families of the children. Inclusion was based solely on a child’s attendance in the class; there were no additional inclusion or exclusion criteria. In situations where both parents were illiterate, a representative from the school visited the homes and provided a verbal explanation of the purpose of the study and obtained verbal consent from the head of the household, which was recorded as an ‘X’ on the consent form in the presence of a witness who countersigned. It was clearly explained verbally and in written form that even after signing the consent form, the parents are free to withdraw their child from this study at any point without giving a reason, and this would have no bearing on the child’s grades in any way. The date of birth of each participating child was recorded and the child was assigned an ID number for the purposes of the study database. Only the head teacher was able to link the name to the ID number.

### 2.4. Study Design

This investigation was a 3-arm 12-month controlled trial, with three data collection points (baseline, T1; midpoint T2; and endpoint, T3). An overview of the design and timeline is provided in Figure 1. Children were assigned to one of three study arms according to their school and class. Study arm 1 (school feeding + MNP) was comprised of children attending the preparatory class in schools A and B. Study arm 2 (school feeding) was comprised of children attending the class 1 in schools A and B. Study arm 3 (no school feeding, control) was comprised of children attending the preparatory class and class 1 in school C.

### 2.5. Intervention and Context

At each of the schools A and B, the preparatory class (*n* = 70) and class 1 (*n* = 70), comprised of boys and girls, served as the intervention classes, receiving the school meal for the entire duration of the study. Although the preparatory class accepts children from the age of 4 years, there is no enforced requirement for children to attend school at all, therefore the age at which they begin and end their schooling is determined by the family and their circumstances. Older children may also enter into the preparatory class or class 1 if they have never attended school before, thus the age range of the children within a class can be broad. In addition, the exact date of birth of a child is sometimes unknown, in which case the year of birth was used to estimate the age. The school lunch was designed by the study nutritionist, based on the cultural norms of the area, the cost and availability of food times. The meal was comprised of rice, ghee and beef, served to the children on trays, with one tray per 4 children. The nutrient intake per child was estimated using Windiets (Robert Gordon University, Aberdeen, UK) and is presented in Table 1. The control group (study arm 3) at school C was comprised of children from the preparatory class and class 1, and was not part of the school meal provision programme. In this school, children can attend informally from age of 3 years in the preparatory class, to sit in with an older sibling or an extended family member. After 5 months, micronutrients were added to the school lunch for children in the preparatory class attending schools A and B (study arm 1). The composition of the Micronutrient Powder (MNP; donated by DSM Nutritional Products ASIA Pacific PTE. Singapore 117440) is given in Table 2. The MNP was added to the lunch just before consumption, such that each sachet of the MNP served about 20 portions of lunches, amounting to an intake of 0.4 g of powder per child. Throughout the study, there were no health restrictions for receiving the meal and all children were provided a meal for the intervention groups. 

Height, weight and mid upper arm circumference were measured at the baseline by trained staff. Weight was measured with the participants being barefoot and minimally clothed, using a standard calibrated digital scale. Height was measured using a stadiometer with participants being barefoot, in the free-standing position. Body Mass Index (BMI) was calculated from weight (kg)/height (m)^2^. Height for age, weight for age and BMI for age z-scores were calculated using the World Health Organization (WHO) AnthroPlus calculator [22], based on the 2006 WHO Child Growth Standards [23].

### 2.6. Primary Outcome Measure

The primary outcome for this trial was the RCPM. The RCPM [24] is a widely used nonverbal intelligence test for children between the ages of 4 and 11 years. The test comprises 36 items in the form of three sets of 12 items, that increase in difficulty. Each item is printed on a single page and consists of a matrix of geometric figures, with one figure missing. At the bottom of the page, six patterns are printed, with one matching the missing figure. Children are asked to choose which of those six alternative figures fits into the missing section of the matrix, with the possible final score ranging from 0 to 36. The psychometric properties of this measure in children are considered good [25], providing an indication of the ability to think clearly and make sense of complexity, and the ability to store and reproduce information. RCPM also has the advantage of being non-verbal, and thus “culture free”, and has been used with children in Pakistan before [26]. The RCPM draws upon the respondent’s ability to find similarities, differences and patterns, which are seen as indicators of executive functioning [27] rather than the respondent’s knowledge or language skills. The RCPM minimizes the need for instructions and specific abilities, such as typical language comprehension or production [28]. The RCPM test was performed at baseline, midpoint and endpoint of the study, and the total scores at each time pointed were used in the analyses. The RCPM test took approximately 20–30 min to complete and was conducted under “exam conditions”, with chairs arranged in rows to prevent copying. All the children were given the instructions in the local language, Pashto, before they proceeded with this test. 

### 2.7. Secondary Outcome Measure

Haemoglobin (g/dL) levels were measured at measured at 6-month and 12-month timepoints of the study using a handheld analyser (HemoCue, Ängelholm, Sweden). Capillary blood samples were collected via a finger prick, using a disposable lancet after cleaning with a 70% ethanol wipe.

### 2.8. Statistical Analyses

Descriptive statistics of the means and standard deviations are presented for each continuous outcome measure and as N (%) for categorial outcomes. Comparisons between trial arms at T1 (i.e., baseline) for continuous outcomes (age, weight, height, mid upper arm circumference (MUAC), weight for age z-score, height for age z-score, BMI for age z-score and RCPM) were undertaken using linear mixed-effects models, with the group modelled as a fixed factor and random intercepts by participants. A Pearson chi-square test of independence was also used to undertake bivariate cross-tabulation comparisons in the number of males/females between the three trial arms. Probability values for the chi-square analysis were calculated using Monte Carlo simulation.

Furthermore, to contrast the magnitude of the changes in the RCPM from T1 (baseline) to T2 (midpoint) and T3 (endpoint) between the three trial arms, linear mixed-effects models, with the group modelled as a fixed factor and random intercepts by participants, were adopted adjusted for covariate baseline scores, age and gender [25]. Finally, to contrast the magnitude of the changes in haemoglobin from T2 to T3 between the three experimental groups, linear mixed models with the group modelled as a fixed factor and random intercepts by participants were adopted, adjusted for T2, age and gender [29]. We undertook these analyses on an intention-to-treat basis and adopted the restricted maximum likelihood method. All analyses for statistical significance were conducted using SPSS v27 (SPSS, IBM Corp, Armonk, NY, USA). For linear mixed models, the mean difference (b), t-value and 95% confidence intervals of the difference are presented (for baseline comparisons, in the interests of conciseness, only *p*-values are presented). Statistical significance for all analyses is accepted at the *p* < 0.05 level.

## 3. Results

### 3.1. Baseline Characteristics

A total of 210 children were recruited into the study, 70 from each of the participating schools. All those invited to participate in the study agreed to do so, thus there were no members of the class that were not enrolled in the study. The number of children completing each data collection time point is provided in Figure 2. The reasons for non-completion included: withdrawal from study because the family moved out of the area (*n =* 26), and temporary absence from school on the day of the test (*n =* 2). Data from children aged less than 4 years at baseline were not included in the RCPM analysis, however, we included children aged 12 years because their literacy levels were comparable to those of their younger classmates.

The baseline characteristics of the participants from each trial arm are described in Table 3. Anthropometric indices of wasting and stunting indicated that the nutritional status of the children attending schools A and B (intervention schools) was worse than that of the children attending school C (control school). Schools A and B had rates of stunting (49%) and wasting (23% and 29%, respectively) compared with school C (3% stunted and 4% wasted). Linear mixed models showed that the RCPM at baseline was significantly greater in the school feeding group compared with the no school feeding group (*p* < 0.001) and school feeding + MNP group (*p* = 0.046), and was also greater in the school feeding + MNP compared with no school feeding (*p* = 0.002).

### 3.2. Effects of Intervention on Cognitive Function, Measured Using RCPM

Adjusted for T1 and other co-variates, improvements in the RCPM scores were significantly greater in the school feeding group at T2 (b = 1.61, (95% CI = 0.71–2.52), t = 3.52, *p* = 0.001) and T3 (b = 1.28, (95% CI = 0.22–2.35), t = 2.38, *p* = 0.019) compared with no school feeding. In addition, adjusted for baseline and other co-variates at T2 (b = 1.63, (95% CI = −0.10–3.37), t = 1.86, *p* = 0.065), there were no significant differences between the school feeding + MNP and no school feeding groups. However, improvements in RCPM scores were significantly greater in the school feeding + MNP group at T3 (b = 2.35, (95% CI = 0.51–4.20), t = 2.53, *p* = 0.013) compared with no school feeding. Finally, improvements in RCPM scores were significantly greater (b = 2.37, (95% CI = 0.67–4.06), t = 2.76, *p* = 0.007) in the school feeding group at T2 compared with school feeding + MNP, although no differences were evident at T3 (b = 0.36, (95% CI = −1.60–2.33), t = 0.37, *p* = 0.72) (Table 4).

### 3.3. Effects of Intervention on Haemoglobin

Mean haemoglobin concentration for the participating children at T2 and T3 were above the cut-off for anaemia for children of <11 mg/dL, used in the recent Pakistan National Nutrition Survey [7] (Table 5). Adjusted for co-variates, there were no significant (*p* > 0.05) alterations in haemoglobin as a function of any of the intervention groups (Table 5).

## 4. Discussion

Although previous studies indicate consistent positive effects of school feeding on energy intake, micronutrient status, school enrolment, and attendance, the impact of school feeding on growth, cognition and academic achievement of school-aged children receiving a meal at the school as part of an SFP compared with non-school-fed children, remain unknown [10,11]. The primary aim of this trial undertaken in early school-age children from a marginalised community in northwest Pakistan was to determine whether receiving a school meal can affect cognitive function, whereas the secondary aim was to explore the effects of the school meal on haemoglobin concentration as an indication of the presence of anaemia. The most common causes of anaemia include nutritional deficiencies, particularly iron deficiency, but may also indicate deficiencies in folate, vitamins B12 and A and parasitic infections. To our knowledge, this trial represents the first study to investigate the influence of providing school lunch alone or in combination with MNP on cognitive ability, in early school age children from the northwest region of Pakistan.

The nutritional status of the children at baseline was worse in the intervention schools (schools A and B), with higher levels of stunting and wasting among the children, compared with the control school (school C). This reflects the nature of the school catchment area, supported by AFPK and AFUK, which seeks to serve the most disadvantaged and marginalised families. Schools A and B have students belonging to Afghan refugee families, displaced people and low paid worker families, whereas school C is in an area where the surrounding population is more settled and living in the area for generations. The higher RCPM scores in the intervention schools at baseline compared with the control school (Table 4) would appear to contradict the hypothesis that nutritional status is positively associated with cognitive performance, however, the age of the children needs to be taken into consideration. The children in the control school were younger than those in the intervention schools at baseline (Table 3), thus their RCPM scores would be expected to be lower [30]. Thus, analysis of the scores in response to the intervention examined the change relative to the baseline. Importantly, we observed significantly greater improvements from baseline to 6 months in the school feeding group and from T1 to T3 in the school feeding + MNP group in relation to the control. Although there is an abundance of evidence from interventions addressing the impact of early childhood (infant and preschool) undernutrition on cognitive ability, there is a paucity of assessment of benefits among school-age children, especially from Pakistan. The findings from this trial for the primary outcome concur with those of Soofi et al. [26]- who also showed significant improvements in both RCPM and Draw-a-Person tests for cognitive function among school age children. However, it should be noted that Soofi et al. [26] included only female children, since the program was targeted at girls from the impoverished districts, and their age-group analysis demonstrated improvements only in children >10 years of age. Thus, the current investigation represents the first to observe improvements in early school-age children. It is interesting to note that at T2, the improvement in the RCPM score in the school feeding group was significantly greater than that of the other intervention group and the control group, since at this time point, there was no difference in the food received by the two intervention groups. We do not have an explanation for this difference at 6 months, however, the gap between RCPM scores of the two intervention groups was closed by 12 months, possibly facilitated by the presence of the additional micronutrients in the school feeding + MNP group.

Micronutrient, vitamin and mineral deficiencies are common in the northwest region of Pakistan [8], and deficiency has been linked to impairments in cognitive function, memory and executive function [31]. As the school meal itself and the MNP included several micronutrients (Table 1 and Table 2), our findings in relation to cognitive function suggest that children in the school feeding and school feeding + MNP groups experienced cognitive benefit as a function of one or more of the micronutrients that were provided. Significant improvements were observed in both groups compared with the control, and there did not appear to be an additional benefit from fortifying the school lunch with MNP. Nevertheless, the findings from this trial show positive responses to school feeding and strongly emphasize the necessity for such interventions in early school-age children. 

A further important observation from this trial is that there was no effect of either school feeding or school feeding + MNP groups on haemoglobin concentrations compared with the control. Though both haemoglobin levels and anaemia have been linked to cognitive performance [32], we did not find patterns for improvement in haemoglobin to explain the observed improvements in cognitive function. It can be speculated that the observed benefits in terms of cognitive function, but not haemoglobin status, could be related to short-term hunger mitigation by the provision of the school meal. It is important to emphasize that micronutrient deficiency at a level not affecting Hb status, such as iodine, vitamin B12, or zinc deficiencies may negatively impact the cognitive abilities of children [27,31]. Though we did not measure the individual biomarkers for these micronutrients, we speculate that the small amount of meat and MNPs served as part of the interventions could have mediated the improvements in cognitive function. Previously, inclusion of animal food, particularly meat [33] (considered to be a rich source of bioavailable micronutrients), and MNP [34], have shown to increase cognitive performance in children. Interestingly, a relatively recent double-blind, controlled clinical trial on children aged 6–9 years, using skim milk 8.8 g powder as ASF for supplementing a micronutrient fortified porridge, in a school meal program in Ghana, found enhanced cognitive function scores in certain domains and accretion of more lean body mass, but no influence on linear growth as compared with a control group, consuming only fortified porridge [35]. The authors attribute the affect to possibly bioactive peptides and additional B12. It is possible that additional animal food could be partially responsible for an improvement in cognitive function in this population subsisting on plant-based diets, as in our study, since we found that at T2, when only lunch (without MNP) was served, showed an improvement in cognitive function over the control group for the children and it could be a more sustainable approach.

### Strengths and Limitations

Overall, the current investigation showed no adverse incidences alongside improvements in cognitive function in the school feeding and school feeding + MNP groups. Therefore, it can be concluded that the provision of a school meal, with or without MNP, is a safe and tolerable approach for the enhancement of cognitive function in early-stage primary school children in northwest Pakistan. A strength of the study was the use of the RCPM, which is non-verbal and independent of the cultural context, allowing for comparison with differing international and socioeconomic environments. In addition, the study was conducted over a 12-month period, such that the baseline and endpoint tests were both conducted at the same point in the academic calendar, thus negating any potential seasonal effects. A limitation of the trial is that the sample was not randomized and that the baseline characteristics of the three schools were not homogeneous. This is reflected in the difference in mean age between the children receiving lunch and lunch with micronutrients (mean age 8.5 and 7.9 years, respectively) and those not receiving the school lunch (mean age 5.1 years) (Table 3). Although age was included as a co-variate, it may be that due to the difference in age, the children in the control group were at a different developmental stage, which may have influenced their potential to improve their RCMT score over the period of the study. In addition, the children at school C were predominantly male (90% boys in study arm 3) compared with schools A and B, where there was a more equal distribution (39% boys in study arm 1 and 54% boys in study arm 2). However, it is unlikely that this would have had a major impact on the longitudinal within-arm comparisons. Another potential limitation of the current trial is that the food was served on shared trays (in accordance with cultural norms), so it is only possible to estimate the quantity of nutrients consumed. In addition, biomarkers for micronutrients such as zinc and iodine were not examined, meaning that it was not possible to explore the relationship between micronutrient status and observed improvements in cognitive function. Therefore, future investigations should seek to explore and perhaps better utilise and exploit the mechanistic pathways of the provision of school meals fortified with micronutrients, in order to improve cognitive function and potentially other health-related outcomes. The intervention was interrupted for 6 weeks during the school summer vacation, when no school meals were provided to the children in arms 1 and 2. Thus, it is possible that the quantity and nutritional composition of food provided to the child at home by the families of children in the school feeding and school feeding + MNP groups on non-designated school days and during school holidays could have affected the impact of these interventions during the study period [36]. 

## 5. Conclusions

The data from this study are important, as they add to the evidence to support the provision of a school meal for improved cognitive ability of school entry-level children from the disadvantaged community of the northwest region in Pakistan. Importantly, this trial was able to demonstrate an improvement in the RCPM that measures general intelligence. The addition of MNP further enhanced the micronutrient content of the meal, and may have mediated the improvements in cognitive function, observed in arm 1 in the second 6-month period of the trial. Currently, there is no SFPs operational at either national or provincial level in Pakistan. Our findings, therefore, support the need for provision of a school meal to improve learning opportunities for children from the most underprivileged communities.

## Figures and Tables

**Figure 1 nutrients-15-01768-f001:**
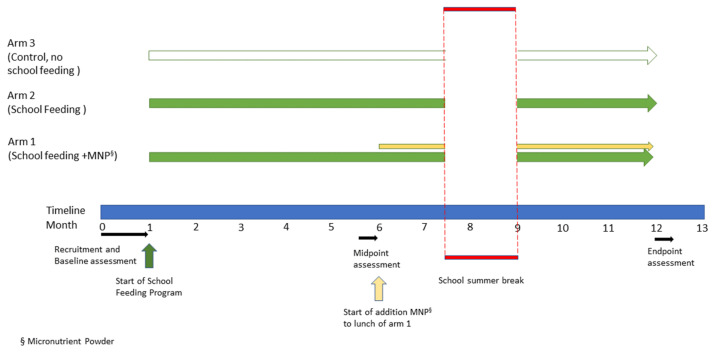
Overview of the study design and timeline.

**Figure 2 nutrients-15-01768-f002:**
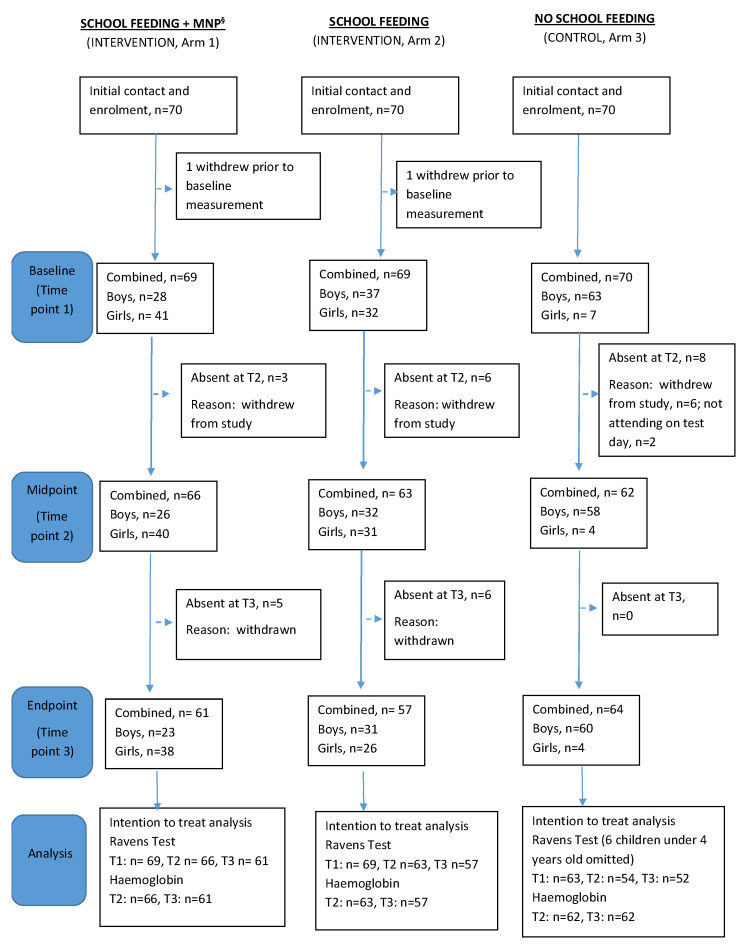
Participant flow diagram showing participant withdrawals at each time point. ^§^ Micronutrient powder.

**Table 1 nutrients-15-01768-t001:** Nutrient content of the school lunch.

Nutrients ^¶^	Amount per Serving	Contribution to Daily Intake Recommendation (%) ^§^
Energy (kcal)	593	27.2–48.4
Fat (g)	38	44.7–110.0
Protein (g)	10	24.4–61.7
Carbohydrates (g)	57	NA
Vitamin A (µg RAE)	0	-
Iron (mg)	1	3.4–7.9
Zinc (mg)	2	11.7–20.8
Calcium (mg)	8	0.2–1.3
Iodine (µg)	3	2.5–3.3
Copper (mg)	1	NA
Folic acid (µg)	39	9.8–19.5

**^¶^** Nutritive value of food was calculated using WinDiets 2016 (Robert Gordon University, UK) food composition data base. One serving of the meal was served to each child and comprised meals cooked with 70 g of brown rice, 28 g beef and 20.5 g sunflower oil. **^§^** Percentages are calculated based on intakes recommended by FAO/WHO for children 4–12 years old [21]. The iron requirement for pre-menarche was considered for girls > 10 years. Iron and zinc recommendations at the lowest level of bioavailability were considered (5% for iron and 15% for zinc, respectively). NA—not available.

**Table 2 nutrients-15-01768-t002:** Composition of the Micronutrient Powder (MNP).

Nutrient	Amount/0.4 g	Nutrient Source	Contribution to Daily Intake Recommendation (%) ^§^
Vit A RE	500 μg	Vitamin A palmitate 250,000 IU/g (beadlet) or Vitamin A acetate 325,000 IU/g (beadlet)	80–125%
Vitamin D3	5 μg	Dry vitamin D3 100,000 IU/g (CWS or beadlet)	100%
Vit. E TE	7 mg	Dry vitamin E acetate 500 IU/g (CWS)	70–140%
Vit. K1	60 μg	Dry vitamin K1 (5% CWS)	109–300%
Vitamin B1	0.9 mg	Thiamine mononitrate	75–150%
Vitamin B2	0.9 mg	Riboflavin fine powder or riboflavin 5 phosphate	69–150-%
Vitamin B6	1 mg	Pyridoxine hydrochloride	77–160%
Vitamin B12	1.8 μg	Cyanocobalamin (1% or 0.1%)	75–150%
Niacin	12 mg	Niacinamide	75–150%
Folic acid	180 μg *	Folic acid	75–150%
Vitamin C	30 mg	Ascorbic acid fine powder	75–100%
Iron	12.5 mg	NaFeEDTA (2.5 mg) + Ferric pyrophosphate micronized (difference) or coated ferrous fumarate	43–99%
Zinc	5.6 mg	Zinc sulphate or Zinc gluconate	33–58%
Copper	0.6 mg	Copper gluconate or Copper sulphate	NA
Iodine	120 μg	Potassium iodide	30.0–60.0

CWS: cold water-soluble. * Equivalent to 300 μg dietary folate equivalent. **^§^** Percentages are calculated based on intakes recommended by the FAO/WHO for children 4–12 years old [21]. The iron requirement for pre-menarche was considered for girls > 10 years. Iron and zinc recommendations at the lowest level of bioavailability were considered (5% for iron and 15% for zinc, respectively). NA—not available.

**Table 3 nutrients-15-01768-t003:** Baseline characteristics of the children that were included in the RCMT analyses.

Baseline (T1) Characteristic	School Feeding (*n* = 69)	School Feeding + MNP (*n* = 69)	No School Feeding (*n* = 63)
Age (years) mean (SD) Age range (years)	8.5 (1.9) 5.0–12.0	7.9 (1.6) 5.0–12.0	5.1 (0.9) 4.0–6.0
Sex, N (%) Male Female	37 (54%) 32 (46%)	28 (41%) 41 (59%)	56 (89%) 7 (11%)
Mass (Kgs), mean (SD)	27.3 (7.08)	24.7 (5.00)	20.6 (2.54) ^¶^
Height (cm), mean (SD)	126.4 (9.76)	122.3 (9.69)	117.8 (7.26)
MUAC (cm), mean (SD)	17.8 (1.64)	18.5 (7.87)	16.1 (1.09)
Weight for age z-score ^§^, mean (SD) Total number wasted (%) % Mild (z-score −1.0 to −1.9) % Moderate (z-score −2.0 to −2.9) % Severe (z-score ≤ −3.0)	0.0 (1.30) 10 (23) 80 20 0	−0.24 (1.32) 17 (29) 65 29 6	0.19 (0.77) ^¶^ 4 (6) 75 25 0
Height for age z-score, mean (SD) Total number stunted (%) % Mild (z=score −1.0 to −1.9) % Moderate (z-score −2.0 to −2.9) % Severe (z-score ≤ −3.0)	−1.05 (1.79) 32 (49) 34 41 25	−0.99 (1.77) 33 (49) 33 77 53	0.78 (1.26) 3 (5) 67 33 0
BMI for age z-score, mean (SD)	0.12 (1.48)	0.08 (1.61)	−0.47 (0.95) ^¶^

^§^ Weight for the age z-score was calculated only for children aged ≤10 years, *n* = 164. ^¶^ One datapoint for mass was removed due to the implausible BMI for the age z-score, as indicated by the AnthroPlus software manual [22]. MUAC is mid upper arm circumference. BMI is body mass index.

**Table 4 nutrients-15-01768-t004:** Means and standard deviations for the Ravens’ Coloured Progressive Matrices (RCPM) at each time point in the intervention and control groups.

	School Feeding			School Feeding + MNP *			No School Feeding		
	*n*	Mean	SD	*n*	Mean	SD	*n*	Mean	SD
RCPM (T1)	69	12.38	5.17	69	10.65	4.89	63	8.62	4.24
RCPM (T2)	63	14.27	5.34	66	11.15	4.62	54	9.33	3.99
RCPM (T3)	57	15.04	6.82	61	13.54	5.15	52	10.08	3.23

The range of possible scores for the RCPM is 0–36. ***** MNP is micronutrient powder.

**Table 5 nutrients-15-01768-t005:** Means and standard deviations for haemoglobin at time points T2 and T3 in the intervention and control groups.

	School Feeding			School Feeding + MNP *			No School Feeding		
	*n*	Mean	SD	*n*	Mean	SD	*n*	Mean	SD
Haemoglobin (g/dL) (T2)	63	11.56	0.77	66	11.42	0.76	62	11.58	0.86
Haemoglobin (g/dL) (T3)	57	11.39	0.84	61	11.45	1.24	62	11.33	1.15

* MNP is micronutrient powder.

## Data Availability

Original data can be obtained on request from the corresponding author.

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
