# Peer review of "School Feeding to Improve Cognitive Performance in Disadvantaged Children: A 3-Arm Parallel Controlled Trial in Northwest Pakistan"

_nutrients, 2023, doi:10.3390/nu15071768_

Round 1
Reviewer 1 Report
Dear Authors,
Thank you for the opportunity to review your manuscript “School Feeding to improve Cognitive Performance in disadvantaged Children: A 3-arm parallel controlled trial in Northwest 3 Pakistan”
Initial thoughts for consideration.
Why was the Raven’s Coloured Progressive Matrices (RCPM) used as the assessment regardless of the fact that it is popular, why is this preferable to other available tests. How has this been validated?
When Hemoglobin samples taken, by whom and what is the device being used. Was there a standard measure of blood taken?
Line 26: “accentuate” may not be the correct word to use in this context.
Line 39: please explain the meaning of poverty traps.
Line 24, there is a need for a reference after the word Children. Please also re punctuate the sentence with respect to ensure correct grammar is used.
Line 46: please explain the Global Hunger index and why it is important in this case??
Line 48:please explain why your definition of malnutrition is.
Line 56: reference ager the word development.
Line 69 ref after the word served.
Line 82: Please explain what is meant by brick-kiln community? Also please explain why this area was chosen for the study as it is not clear.
Line 86: what is basic conditions as there is no way to determine what is meant by this term.
Line 160: The meal given has not been explained,. Is there a rationale for this meal, why was this level utilized and who designed the diet in this fashion and could this have been achieved with other foods being utilized. It is interesting the sunflower oil is used in preference to extra virgin olive oil. Was this due to cons or the density of fat required?? Did a nutritionist design the diet??
Line 190: The micronutrient used is not explained. Who chose this, why? It is a specific brand or simply designed for this trial? Has this formulation been validated or a recommendation for a health statis with respect to children?
Results table 3,4 & 5, it would be very interesting to also have the results as both male, female and total as it would determine outcomes in relation to sex and how much nutrient value is required for the different genders. This is not clear in the results.
Discussion: You state previous studies, however they are not referenced and not explained how they relate to the current study.
Conclusion: with the inability to measure the impact of cognitive improvements, the only outcome is indicative and may indicate the need for school feeding , however, this is not an outcome of this study.
Author Response
Many thanks for your helpful comments. Our responses are in the attached document. Apologies if the line numbers have shifted during the re-formatting process.

Reviewer 2 Report
nutrients-2274661
School Feeding to improve Cognitive Performance in disadvantaged Children: A 3-arm parallel controlled trial in Northwest Pakistan.
Dear authors,
Your work is very interesting and important. However, there are parts of the results that are not quite plausible. You have not discussed what could be the reason that in the TP2 assessment there are no differences in intelligence between the food+supplements group and the control group, when up to that point it is the same intervention as in the food-only group. I find it very striking that it is in the food-only intervention group that the intelligence is higher, when in some way, the supplements are made to be more assimilable by the organism and therefore, we can absorb them to a greater extent. So, if you attribute the benefits to better nutrition, this should be more complete in the supplemented group if we consider that vitamin supplements are usually more easily absorbed. This is something you should discuss further in the discussion.
Here are some comments:
Abstract: Add the number of participants. Add CI or p-value in the result
Introduction: It is not necessary to include rationale and aims sub-paragraphs, tPopulation: indicar claramente cuántos niños participaron finalmente
Primary outcome: Include possible scores and their interpretations.
Study variables: add a sub-section on "other variables or covariates" and describe the adjustment variables of the linear regressions.
Statistical analysis: what is MUAC, have you assessed the normality of continuous variables, indicate and describe.
Figure 2: add baseline assessment
Results: when and how did you calculate the wasted and stunted rates? Add in methodology.
Results Line 240: The results suggest better cognitive function in schools a and b compared to c, despite the higher % of malnutrition in these schools. This is somewhat contradictory to your initial hypotheses, is it possible that this is due to the age difference between the schools? Discuss this.
Results Lines 276-278: It is very curious that there is no higher score in the RCPM of the group intervened with food + supplements versus the control group in TP2, but there is a higher score in the group intervened only with food. This is striking because the use of supplements is not included until TP2, and therefore, the intervention of schools a and b would be exactly the same and the baseline data are very similar between them. This analysis needs to be revisited and/or reasoned out in the discussion.
Results table 4 and 5: It is necessary to change the current table 4 with the data from the RCPM regression models. The same for table 5. In the results you talk about the models adjusted for different covariates, but in tables 4 and 5 you only show the mean and SD data.
Discussion: Describe SFP
Discussion Lines 314-315: Do you mean that school c was in a less disadvantaged area than schools a and b and that is why there is a lower % of stunted? Clarify
Discussion Lines 318-327. I understand this information. But how do you explain that there are no significant differences in the RCPM between the lunch+MNP group vs. the control group at TP2 when there are significant differences between the lunch group vs. the control group? Taking into account that until TP2 the intervention is the same in schools a and b.
Discussion Line 336: That there are higher cognitive function scores in the food-only group versus the food + supplements group in TP2, when the intervention is the same, somehow indicates that the populations are not comparable, there is some baseline characteristic that must be causing the same intervention to work better in one school than in another with supposedly similar characteristics. Provide an explanation.
Discussion: Lines 374-375: The mean ages that you indicate are not those in table 3. Revise
Discussion: Add in the limitations that the sample was not randomised and that the baseline characteristics of the 3 schools were not homogeneous.
Conclusion: Lines 398-399: In the results you have not differentiated between the two components. You should do so if you include them in the conclusion.
Thank you
Author Response
Thank you for your helpful comments. Our responses are in the attached document. Apologies if the line numbers have shifted in the reformatting process.

Reviewer 3 Report
I read the manuscript entitled "School Feeding to improve Cognitive Performance in disadvantaged Children: A 3-arm parallel controlled trial in Northwest Pakistan" with great interest. The study presented by the authors is very interesting and needed. However, the results obtained are not too surprising, proving only the benefits of school meals offered to children. The study is not without significant limitations, most of which the authors pointed out. One particularly important limitation of the study presented is the significant age difference in children between the groups, as well as the several-week hiatus in the study due to the summer vacation.
The primary conclusion of the study is the benefit of introducing a school feeding program for children, while no confirmation was obtained of the clear benefits of enriching the school meal with a powdered mixture of vitamins and minerals, and this should clearly and definitely be included in the authors' conclusion.
Below are my detailed comments on the text of the manuscript:
1. first of all, I ask that all abbreviations introduced and used be explained at the first place of their use. In addition, this is particularly troublesome in the case of abbreviations introduced in tables and figures - the absence of an explanation of these abbreviations well in advance makes it much more difficult or even impossible to properly read the data presented.
2. I would ask to standardize throughout the text introduced designations for groups and time points. In some places there are differences in these designations, which raises unnecessary doubts.
3. Abstract section - I would ask for a clearer and more unambiguous description of the results obtained; no results regarding hemoglobin determination; no inference regarding the effects/impacts of MNP application.
4. Please clarify the authors' use of the term "child wasting" (first included in line 49) used repeatedly throughout the manuscript. It is incomprehensible to me, I don't know what it refers to and what exactly it means.
5. The extraction of two subsections (1.1 Rationale and 1.2 Aims) from the Introduction section seems unnecessary.
6. Materials and Methods section:
- please include the timing of the study and separate the inclusion and exclusion criteria for the study
- if possible, please supplement the text with information on how many meals the children participating in the study consumed outside of school, or what percentage of the meal offered at school covered the children's daily energy and nutrient needs
- Table 1 shows the composition of the school meal - please explain why only certain minerals and vitamins were included?
- how was the MNP powder administered to the children in the meal? We have information: "The MNP was added to the lunch just before consumption such that each sachet of the MNP served about 20 portions of lunch amounting to an intake of 0.4 g of powder per child." (lines 154-156), but there is no information on the method of administration - was it mixed with the meal? In addition, I understand that one meal (on one tray) was consumed by four children, so it is difficult to say that each child consumed the same amount of this powder
- Table 2 gives the composition of the powder - I would ask you to complete the information on what percentage it covered the children's daily requirement for each vitamin and mineral
7. Results section:
- in Figure 2, the symbol "*" was introduced in some blocks without its explanation - please supplement this
- in subsection 3.2, in the description of the results, the symbols "TR1", "TR2" and "TR3" were introduced without their previous explanation. In all likelihood, you can guess their meaning, but please introduce precise explanations
- in Table 5, it would be worthwhile to insert symbols identifying the presence of statistically significant differences, which would definitely make it easier to read the posted data.
8. The section on limitations of the study should be supplemented with the way the school meal was served to the children ("The school lunch was comprised of rice, ghee and beef, served to the children on trays, with one tray per 4 children." Lines 145-146), because in the way the meal was presented to the children, it is difficult to really say that all the children consumed the same amount of food and therefore energy and individual nutrients.
9. The conclusion section should be supplemented with a conclusion regarding the small effect of the powder used on the cognitive functions of the children studied.
10. The References section in its entirety should be adapted to the requirements of the journal.
Author Response
Thank you for your helpful comments. Our responses are in the attached document. Apologies if the line numbers have shifted in the reformatting process

Round 2
Reviewer 2 Report
Dear authors,
Thank you for trying to include all the changes I suggested. However, I am still concerned that the tables show descriptive data, when your conclusions are based on the results of multivariate analyses. I understand your response, but I still believe that the results of the multivariate analysis should be shown in the tables. The results of the fitted models would be sufficient.
Author Response
With great respect we do not agree with this request, as said information is already included within the main body of the text, in the results section alongside the interpretation of the findings e.g.
Adjusted for T1 and other co-variates improvements in RCPM scores were
significantly greater in the school feeding group at T2 (b = 1.61, (95% CI = 0.71 – 2.52), 291
t = 3.52, P=0.001) and T3 (b = 1.28, (95% CI = 0.22 – 2.35), t = 2.38, P=0.019) compared
to no school feeding.
Unfortunately, we believe that the requested approach in placing the model information in tables would not be appropriate as it would severely compromise the clarity of the results section to the Nutrients reader. We feel strongly that this alteration would be scientifically irregular in the context of the literature as all of our previously published Nutritional trials have adopted this current approach (see below) and would also contravene the general scientific and Nutrients journal specific guidance for results sections to '’provide a concise and precise description of the experimental results, their interpretation, as well as the experimental conclusions that can be drawn’'.
https://www.mdpi.com/2072-6643/14/8/1657
https://www.mdpi.com/1660-4601/19/9/5317
Reviewer 3 Report
I thank the authors for taking my comments into account in the preparation of the revised version of the manuscript.
Please just add an explanation of the abbreviation "MNP" under Figure 1.
In addition, adding "T1-T3" to the caption of figure 2 did nothing, it's still hard to figure out what these designations mean without reading the text. Also missing is the "ITT" explanation.
Author Response
An explanation of MNP has been added to figures 1 and 2.
ITT has been written out in full in Figure 2.
"T1-T3" has been removed from figure 2 and also replace with Time point 1, Time point 2 etc in the blue boxes.